# Plumeriapropionics A–E, Carboxyl-Substituted Phenylpropionic Acid Derivatives with Anti-Inflammatory Activity from *Plumeria rubra* L.

**DOI:** 10.3390/molecules29010115

**Published:** 2023-12-24

**Authors:** Xueming Zhou, Minlin Gan, Meizhu Wu, Ting Zheng, Chuluunbaatar Enkhchimeg, Haixiang Li, Shuo Feng, Jingqi Zhou, Xinming Song

**Affiliations:** 1Key Laboratory of Tropical Medicinal Resource Chemistry of Ministry of Education Hainan Normal University, Haikou 571158, China; xueming2009211@126.com (X.Z.); 13876509861@163.com (M.G.); meizhu0812@163.com (M.W.); zhengt0311@163.com (T.Z.); linfangxia@126.com (C.E.); 18976885896@163.com (H.L.); ghhvzfbn@163.com (S.F.); zjq19529746745@126.com (J.Z.); 2Key Laboratory of Tropical Medicinal Plant Chemistry of Hainan Province, College of Chemistry and Chemical Engineering, Hainan Normal University, Haikou 571158, China

**Keywords:** *Plumeria rubra*, plumeriapropionics, carboxyl-substituted phenylpropionic acid, anti-inflammatory activity

## Abstract

Five rare carboxyl-substituted phenylpropionic acid derivatives, plumeriapropionics A–E (**1**–**5**), together with one known analog, cerberic acid B (**6**), were isolated from flowers of *Plumeria rubra* L. Their structures were elucidated using comprehensive spectroscopic methods. To date, only one compound of this structural type has been reported. The inhibitory activities of compounds **1**–**6** against nitric oxide (NO) production induced by lipopolysaccharide (LPS) were evaluated in vitro using mouse macrophage RAW264.7 cells. Compounds **1**–**6** showed remarkable inhibitory activities on NO production, with IC_50_ values in the range of 6.52 ± 0.23 to 35.68 ± 0.17 µM. These results indicate that the discovery of carboxyl-substituted phenylpropionic acid derivatives from the flowers of *P. rubra*, which show significant anti-inflammatory properties, could be of great importance for the research and development of novel natural anti-inflammatory agents.

## 1. Introduction

The genus *Plumeria*, belonging to the Apocynaceae family, consists of approximately ten species native to tropical America, which are mainly distributed in tropical and subtropical regions of Asia [1]. There is one species and one variant in China, mainly growing in Fujian, Guangdong, Guangxi, Yunnan and Hainan provinces [2]. Previous chemical investigations on the plants from the genus *Plumeria* have led to the isolation and identification of a variety of natural products, including iridoids [3,4,5], triterpenoids [4,5,6,7,8], ferulic acids [9], and flavones [5,10], which display various biological activities, such as anti-diabetic [5], anti-tumor [3], and anti-inflammatory [11,12]. Among the genus *Plumeria*, *P. rubra* L. is a deciduous shrub or small tree widely planted in the south of China as an ornamental shrub. The flowers of *P. rubra* are often used in folk medicine for the treatment of various diseases, such as enteritis, acute bronchitis, bacillary dysentery, and infectious hepatitis [2].

Our preliminary experimental results showed that the 75% ethanol extract of the flowers of *P. rubra* showed an inhibitory effect against NO production induced by LPS in mouse macrophage RAW 264.7 cells with an IC_50_ value of 22.89 ± 0.16 µg/mL in vitro. In order to deeply explore the enormous potential of China’s unique tropical medicinal plants in treating and preventing major human diseases, a chemical investigation on the flowers of *P. rubra* was carried out. Bioassay-guided fractionation of the bioactive extract led to the identification of five new carboxyl-substituted phenylpropionic acid derivatives named plumeriapropionics A–E (**1**–**5**), together with one known analogue, cerberic acid B (**6**) (Figure 1). Structure elucidations of these compounds were clarified by comprehensive spectroscopic analyses, including NMR spectral data, HR-ESI-MS data, IR, optical rotations, and comparisons with the spectral data reported in the literature. Compounds **1**–**6** are a kind of rare carboxyl-substituted phenylpropionic acid derivatives. Although phenylpropanoid is a common component in plants, carboxyl-substituted phenylpropanoid is very rare in the plant kingdom. So far, only one compound of this structural type has been reported. To explore the potential of these isolated carboxyl-substituted phenylpropionic acid derivatives for the development of anti-inflammatory drugs, compounds **1**–**6** were evaluated for their anti-inflammatory activities in vitro. Herein, we will report the isolation, identification, and pharmacological activity of these compounds.

## 2. Results and Discussion

### 2.1. Phytochemical Investigation

The dried flowers of *P. rubra* were extracted with 75% ethanol, and the extract was subjected to column chromatography over a silica gel, octadecylsilyl silica gel (ODS), Sephadex LH-20, as well as semi-preparative high-pressure liquid chromatography (HPLC) to yield five new carboxyl-substituted phenylpropionic acid derivatives, plumeriapropionics A–E (**1**–**5**), and one known analog, cerberic acid B (**6**). Compounds **1**–**6** are rare natural products isolated from the genus *Plumeria* for the first time.

Compound **1** was obtained as a white amorphous powder. The molecular formula of C_18_H_16_O_6_ (11 degrees of unsaturation) was deduced by high-resolution electrospray ionization mass spectrometry (HRESIMS) combined with ^1^H- and ^13^C-NMR data (Table 1). The ^1^H/^13^C-NMR and 135DEPT data revealed one 1,3-disubstituted benzene ring *δ*_H_ 8.00 (1H, br s, H-2), 7.82 (1H, dd, *J* = 7.8, 1.2 Hz, H-4), 7.64 (1H, d, *J* = 7.2 Hz, H-6), 7.45 (1H, dd, *J* = 7.8, 7.2 Hz, H-5) and *δ*_C_ 137.3 (C-1), 134.5 (C-6), 130.4 (C-2), 129.7 (C-3), 128.9 (C-5), 127.8 (C-4); one benzoyl group *δ*_H_ 7.94 (2H, m, H-2′,6′), 7.65 (1H, m, H-4′), 7.48 (2H, m, H-3′,5′) and *δ*_C_ 165.2 (C-7′), 133.8 (C-4′), 130.9 (C-1′), 129.4 (C-2′,6′), 128.9 (C-3′,5′); one oxygenated methine proton *δ*_H_ 5.36 (1H, dd, *J* = 7.8, 4.2 Hz, H-8) and *δ*_C_ 73.1 (C-8); one methoxy group *δ*_H_ 3.83 (3H, s, 10-OMe), *δ*_C_ 52.2 (10-OMe); one methylene group *δ*_H_ 3.38 (1H, dd, *J* = 14.4, 4.2 Hz, H-7b), 3.30 (1H, dd, *J* = 14.4, 7.8 Hz, H-7a) and *δ*_C_ 36.4 (C-7); two carboxyl carbons *δ*_C_ 170.4 (C-9) and 166.3 (C-10). The ^1^H-^1^H COSY correlation of H-7/H-8 combined with the HMBC correlations from H-7 and H-8 to C-9 indicated the presence of fragments from C-7 to C-9 (Figure 2). The HMBC correlations of H-2/4 with C-10 and H-7 with C-1/2/6 suggested the linkages of C-3–C-10 and C-7–C-1. The HMBC correlations of H-8 with C-7′ indicated that the benzoyl group was linked at C-8 by an ester bond. The location of the methoxy group at C-10 was confirmed by the HMBC correlation between 10-OMe and C-10. Therefore, the planar structure of **1** was determined. Owing to a putative shared biosynthesis pathway with **1** and **6**, the absolute configuration of **1** was tentatively assigned as 8*R*-form. To substantiate the aforementioned conclusion, the ECD spectra of **1** and **6** were analyzed, revealing comparable characteristics within the 200–400 nm range (see Appendix A). Thus, compound **1** was identified as a rare carboxyl-substituted phenylpropionic acid. We named compound **1** plumeriapropionic A.

Compound **2** was also obtained as white amorphous powder and determined to be C_18_H_16_O_6_ by HRESIMS spectrum (*m*/*z* 327.0878 [M-H]^−^; calcd for C_18_H_15_O_6_, 327.0874), corresponding to 11 degrees of unsaturation. The first preliminary investigation of its ^1^H and ^13^C NMR showed that **2** was closely related to **1**. The differences in chemical shift were at the methoxy group (*δ*_H_ 3.83 for **1** vs. 3.66 for **2**). The location of the methoxy group at C-9 was confirmed by the HMBC correlations from 9-OMe to C-9. This indicates that the methoxy group is substituted at C-9 instead of C-10 in compound **2.** Detailed analysis of 2D NMR (HSQC, ^1^H-^1^H COSY, and HMBC) spectra confirmed that the remaining parts of the molecule were identical to those observed in compound **1**. Compound **2** has a similar optical rotation value to compounds **1** and **6**. The ECD spectra of **2** and **6** showed similar features in the 200–400 nm range (Appendix A). Therefore, the stereostructure of C-8 was also assigned as the *R*-form. Thus, compound **2** was also identified as a rare carboxyl-substituted phenylpropionic acid. We named compound **2** plumeriapropionic B.

The molecular formula of **3** was determined as C_12_H_14_O_5_ based on the HRESIMS (*m*/*z* 239.0911 [M + H]^+^; calcd 239.0914). Its ^1^H/^13^C-NMR data (Table 2) closely resembled those of cerberic acid B (**6**), except for the presence of two methoxy group signals (*δ*_H/C_ 3.84/52.1 and 3.62/52.1) in **3**. The location of the two methoxy groups at C-9 and C-10 was confirmed by the HMBC correlations from 9-OMe to C-9 and from 10-OMe to C-10. Detailed analysis of 2D NMR (HSQC, ^1^H-^1^H COSY, and HMBC) spectra confirmed that the other parts of the molecule were the same as those of **6**. Compound **3** has the 8*R*-form based upon the positive optical rotation of **3** and the reported similar compounds in the literature [13,14]. It was also confirmed by the ECD spectra of **1** and **6**. The ECD spectra of **3** and **6** showed similar features in the 200–400 nm range (Appendix A). Thus, compound **3** was identified as a rare carboxyl-substituted phenylpropionic acid and named plumeriapropionic C.

Compound **4** was also isolated as a white amorphous powder. Its molecular formula was determined to be C_11_H_12_O_5_ by HRESIMS data at *m*/*z* 223.2028 (calcd 223.2026 for C_12_H_11_O_5_). The ^1^H^/13^C NMR data closely resemble that of **3** except for the absence of a methoxy group signal (*δ*_H/C_ 3.62/51.5). It also can be confirmed by comparing the chemical shift of **4** with compound **3** at C-9 (*δ*_C_ 173.8 for **3** vs. 175.6 for **4**). Detailed analysis of 2D NMR (HSQC, ^1^H-^1^H COSY, and HMBC) spectra confirmed that the other parts of the molecule were the same as those of **3**. Owing to a putative shared biosynthesis pathway with **1**–**3** and **6**, the absolute configuration of **4** was tentatively assigned as the 8*R*-form. The ECD spectra of **4** and **6** showed similar features in the 200–400 nm range (Appendix A). The absolute configuration of **4** was also determined as the 8*R*-form based on the positive optical rotation. Thus, compound **4** was also identified as a rare carboxyl-substituted phenylpropionic acid and named plumeriapropionic D.

Compound **5** was also obtained as a white amorphous powder. Its molecular formula of C_11_H_12_O_5_ was determined by HRESIMS data at *m*/*z* 223.2021 (calcd 223.2026 for C_12_H_11_O_5_). Its ^1^H and ^13^C-NMR data also closely resembled those of compound **3** except for the absence of a methoxy signal (*δ*_H/C_ 3.84/52.1) in **5**. This was confirmed by comparing the chemical shift at C-10 of compound **5** with that of compound **3** (*δ*_C_ 166.4 for **3** vs. 167.6 for **5**). The absolute configuration of **5** was also determined as the 8*R*-form based on the positive optical rotation and the ECD spectra of **5** and **6** (Appendix A). Thus, compound **5** was also identified as a rare carboxyl-substituted phenylpropionic acid and named plumeriapropionic E.

In addition, the known carboxyl-substituted phenylpropionic acid **6** was isolated and identified as cerberic acid B [13] by comparing the experimental spectral data with the reported spectra data in the literature.

### 2.2. Anti-Inflammatory Activity

All carboxyl-substituted phenylpropionic acids **1**–**6** were evaluated for their anti-inflammatory effects by testing their inhibitory activities against NO production by LPS in mouse macrophage RAW 264.7 cells in vitro. The MTT assay was used for measuring the cytotoxic activities of compounds **1**–**6** against mouse macrophage RAW 264.7 cells. The results are shown in Table 3. Compounds **2**, **5**, and **6** exhibited especially potent inhibitory activities against NO production, with IC_50_ values of 6.52 ± 0.23, 7.13 ± 0.16 and 6.68 ± 0.22, respectively. Compounds **1**, **3**, and **4** showed weaker activities than **2**, **5**, and **6**. These results suggest that the carboxylic group at C-3 can be important for anti-inflammatory activity. In addition, no cytotoxicity was observed in the macrophage RAW 264.7 cells treated with compounds **1**–**6** (cell viability > 95%)

## 3. Materials and Methods

### 3.1. General Experimental Procedures

Optical rotations of **1**–**6** were measured on a JASCO P-1020 digital polari meter (Jasco Corp., Tokyo, Japan). The NMR spectra of compounds **3**–**6** were recorded on a Bruker AV spectrometer (400 MHz for ^1^H and 100 MHz for ^13^C, Bruker Corp., Karlsruhe, Germany). The NMR spectra of compounds **1** and **2** were recorded on a JEOL JEM-ECP NMR spectrometer (600 MHz for ^1^H and 150 MHz for ^13^C, JEOL, Tokyo, Japan). HRESIMS spectra of compounds **1**–**5** were measured on a Q-TOF Ultima Global GAA076 LC mass spectrometer (Waters Corp., Milford, MA, USA). CD spectra were recorded on a MOS-450 spectrometer. Semi-preparative HPLC was performed on an Agilent 1260 LC (Agilent Corp., Santa Clara, CA, USA) series with a DAD detector using an Agilent Eclipse XDB-C_18_ (Agilent Corp., Santa Clara, CA, USA) column (9.4 × 250 mm, 5 μm). Silica gel ODS and Sephadex LH-20 (Qing Dao Hai Yang Chemical Group Co., Qingdao, China) were used for open-column chromatography (CC). Precoated silica gel plates (Yan Tai Zi Fu Chemical Group Co., Yan Tai, China; G60, F-254) were utilized for thin-layer chromatography (TLC).

### 3.2. Plant Material

The flowers of *P. rubra* (Apocynaceae) were collected from Haikou City, Hainan Province, China, in April 2022 and were authenticated by Professor Yu-Kai Chen (School of Hainan Normal University, Hainan, China). The specimens (No JDH20220428) were deposited at the Key Laboratory of Tropical Medicinal Resource Chemistry of the Ministry of Education, Hainan Normal University (Hainan, China).

### 3.3. Extraction and Isolation

The air-dried flowers of *P. rubra* (5.3 kg) were extracted with 75% EtOH (4 × 20 L) at room temperature. A dark brown crude extract (0.66 kg) was obtained after concentration in vacuo to remove most of the EtOH. The EtOH extract (87 g) was subjected to silica gel CC (100–200 mesh) eluted by petroleum ether/ethyl acetate (100:0 to 0:100, *v*/*v*) to afford nine major fractions (Fr. 1–9). Fr. 3 exhibited an inhibitory effect against NO production induced by LPS in mouse macrophage RAW 264.7 cells. Therefore, systematic separation and purification were carried out against this fraction. The Fr. 3 (32 g) was separated using a silica gel column and eluted with gradient mixtures of petroleum ether/ethyl acetate (10:1 to 1:1, *v*/*v*) to obtain five fractions (Fr. 3-1–3-5). Fr. 3-2 (4.8 g) was purified on Sephadex LH-20 (CHCl_3_:MeOH, 1:1) and further isolated with semi-preparative HPLC (RP C_18_ column, 2.5 mL/min, detected at 210, 230, 254, and 280 nm, CH_3_CN/H_2_O, 30:70 *v*/*v*) to obtain **1** (12 mg), **2** (15 mg), and **3** (21 mg). Fraction Fr. 3-3 (5.2 g) was subjected to ODS CC and eluted with MeOH/H_2_O (25:75), and further isolated using semi-preparative HPLC (RP C_18_ column, 2.5 mL/min, detected at 210, 230, 254, and 280 nm, CH_3_CN/H_2_O, 25:75 *v*/*v*) to obtain **4** (5 mg) and **5** (25 mg). Fr. 3-4 (5.6 g) was repeatedly purified with ODS CC and eluted with MeOH/H_2_O (15:85) to obtain compound **6** (58 mg).

*Plumeriapropionic A* (**1**): white amorphous powder; [*α*]^25^_D_ + 18.4 (*c* 0.10, MeOH); CD (*c* 1.0 × 10^−4^, MeOH) *λ*_max_ (Δ*ε*) 288 (3.23), 262 (2.83), 250 (−6.69), 240 (15.99), 223 (26.01); IR (KBr) *ν*_max_ 3326, 1741, 1708 and 1681 cm; ^1^H-NMR (600 MHz, DMSO-*d*_6_) and ^13^C-NMR (150 MHz, DMSO-*d*_6_), see Table 1; HRESIMS *m*/*z* 327.0876 (calcd for C_18_H_15_O_6_, 327.0874).

*Plumeriapropionic B* (**2**): white amorphous powder; [*α*]^25^_D_ + 22.1 (*c* 0.10, MeOH); CD (*c* 1.0 × 10^−4^, MeOH) *λ*_max_ (Δ*ε*) 283 (3.36), 260 (2.92), 248 (−4.73), 237 (25.44), 219 (58.31); IR (KBr) *ν*_max_ 3328, 1742, 1706 and 1685 cm; ^1^H-NMR (600 MHz, DMSO-*d*_6_) and ^13^C-NMR (150 MHz, DMSO-*d*_6_), see Table 1; HRESIMS *m*/*z* 327.0878 (calcd for C_18_H_15_O_6_, 327.0874).

*Plumeriapropionic C* (**3**): white amorphous powder; [*α*]^25^_D_ + 13.2 (*c* 1.0, MeOH); CD (*c* 1.0 × 10^−4^, MeOH) *λ*_max_ (Δ*ε*) 264 (1.02), 253 (−2.18), 238 (8.38), 218 (6.81); IR (KBr) *ν*_max_ 3334, 1744, 1738 and 1680 cm; ^1^H-NMR (400 MHz, DMSO-*d*_6_) and ^13^C-NMR (100 MHz, DMSO-*d*_6_), see Table 2; HRESIMS *m*/*z* 239.0911 (calcd 239.0914 for C_12_H_15_O_5_).

*Plumeriapropionic D* (**4**): white amorphous powder; [*α*]^25^_D_ + 11.1 (*c* 1.0, MeOH); CD (*c* 1.0 × 10^−4^, MeOH) *λ*_max_ (Δ*ε*) 284 (5.51), 260 (3.57), 248 (−9.26), 238 (12.54), 223 (23.43); IR (KBr) *ν*_max_ 3336, 1743, 1682 and 1595 cm; ^1^H-NMR (400 MHz, DMSO-*d*_6_) and ^13^C-NMR (100 MHz, DMSO-*d*_6_), see Table 2; HRESIMS *m*/*z* 223.2028 (calcd 223.2026 for C_11_H_11_O_5_).

*Plumeriapropionic E* (**5**): white amorphous powder; [*α*]^25^_D_ + 12.3 (*c* 1.0, MeOH); CD (*c* 1.0 × 10^−4^, MeOH) *λ*_max_ (Δ*ε*) 287 (5.31), 264 (6.90), 253 (−0.30), 242 (4.75), 233 (−8.16), 220 (24.19); IR (KBr) *ν*_max_ 3335, 1737, 1681 and 1597 cm; ^1^H-NMR (400 MHz, DMSO-*d*_6_) and ^13^C-NMR (100 MHz, DMSO-*d*_6_) see Table 2; HRESIMS *m*/*z* 223.2021 (calcd 223.2026 for C_11_H_11_O_5_).

*Cerberic acid B* (**6**): white amorphous powder; [*α*]^25^_D_ + 9.8 (*c* 1.0, MeOH); CD (*c* 1.0 × 10^−4^, MeOH) *λ*_max_ (Δ*ε*) 285 (10.54), 261 (12.80), 252 (0.57), 239 (44.49), 220 (50.97), 206 (−38.00).

### 3.4. Anti-Inflammatory Bioassays

All isolated compounds **1**–**6** were evaluated for their inhibition of NO production in RAW264.7 cells activated by LPS using the Griess assay with hydrocortisone as a positive control [15,16]. All experiments were carried out in triplicate, and each experiment was repeated three times. Data analysis was carried out using SPSS statistical package version 22.0 (SPSS, Inc., Chicago, IL, USA). The IC_50_ values (the concentration of drug necessary to induce 50% inhibition) were measured for all of the tested drugs using the Probit test in SPSS software.

## 4. Conclusions

In this study, the phytochemisty study on the flowers of *P. rubra* was carried out and led to the isolation of five rare carboxyl-substituted phenylpropionic acids, plumeriapropionics A–E (**1**–**5**), together with one known analog, cerberic acid B (**6**). Carboxyl-substituted phenylpropanoid is very rare in the plant kingdom. So far, only one compound of this structural type has been reported. The discovery of five rare carboxyl-substituted phenylpropionic acids **1**–**5** can help to extend the phytochemical knowledge of the genus *Plumeria*. All isolated compounds were investigated for their anti-inflammatory effects and were proven to be useful. Compounds **2**, **5**, and **6** showed notable inhibitory effects against NO production. These isolated carboxyl-substituted phenylpropionic acids, with potent inhibitory activities on NO production, could be used for the development of new anti-inflammatory agents.

## Figures and Tables

**Figure 1 molecules-29-00115-f001:**
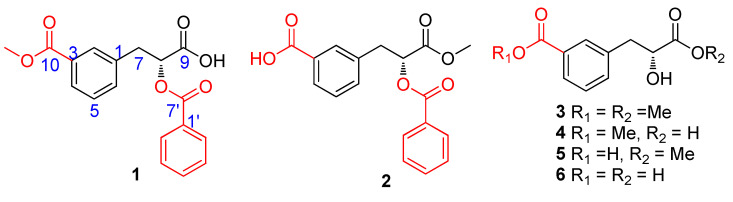
Structures of compounds **1**–**6** from *P. rubra*.

**Figure 2 molecules-29-00115-f002:**
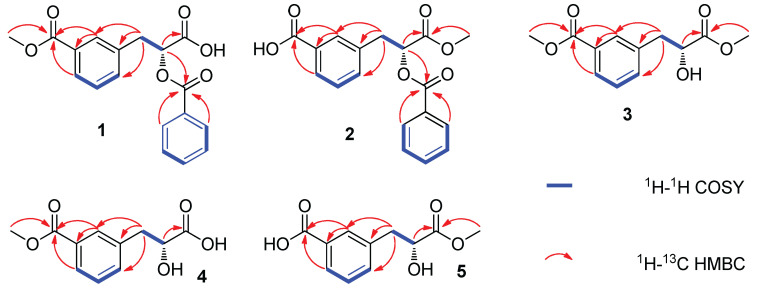
Key HMBC and ^1^H-^1^H COSY correlations of **1**–**5**.

**Table 1 molecules-29-00115-t001:** ^1^H and ^13^C NMR spectral data of compounds **1** and **2**.

Position	1 ^a^		2 ^a^	
	*δ*_H_, Mult, (*J* in Hz)	*δ* _C_	*δ*_H_, Mult, (*J* in Hz)	*δ* _C_
1	-	137.3	-	136.5
2	8.00 (br, s)	130.4	7.96 (s)	130.6
3	-	129.7	-	128.7
4	7.82 (dd, 7.8, 1.2)	127.8	7.83 (d, 7.8)	128.0
5	7.45 (dd, 7.8, 7.2)	128.9	7.44 (dd, 7.8, 7.8)	128.7
6	7.64 (d, 7.2)	134.5	7.57 (d, 7.8)	134.0
7	3.38 (dd, 14.4, 4.2) 3.30 (dd, 14.4, 7.8)	36.4	3.37 (dd, 14.4, 4.8) 3.31 (dd, 14.4, 7.8)	36.4
8	5.36 (dd, 7.8, 4.2)	73.1	5.46 (dd, 7.8, 4.8)	72.9
9	-	170.4	-	169.5
10	-	165.2	-	165.1
9-OMe	-	-	3.66	52.3
10-OMe	3.83 (s)	52.2	-	-
1′	-	130.9	-	130.9
2′,6′	7.94 (m)	129.4	7.95 (m)	129.4
3′,5′	7.48 (m)	128.9	7.49 (m)	129.0
4′	7.65 (m)	133.8	7.67 (m)	134.0
7′	-	165.2	-	165.1

^a^ measured in DMSO-*d*_6_ at 600 MHz.

**Table 2 molecules-29-00115-t002:** ^1^H and ^13^C NMR spectral data of compounds **3**–**5**.

Position	3 ^a^		4 ^a^		5 ^a^	
	*δ*_H_, Mult, (*J* in Hz)	*δ* _C_	*δ*_H_, Mult, (*J* in Hz)	*δ* _C_	*δ*_H_, Mult, (*J* in Hz)	*δ* _C_
1	-	138.5	-	139.9	-	138.3
2	7.83 (s)	130.2	7.84 (s)	130.7	7.81 (s)	130.5
3	-	129.5	-	129.8	-	130.7
4	7.81 (d, 8.0)	127.2	7.79 (d, 7.6)	127.4	7.80 (d, 8.0)	127.5
5	7.42 (dd, 8.0, 8.0)	128.5	7.41 (dd, 7.6, 7.6)	128.8	7.39 (dd, 8.0, 7.6)	128.4
6	7.49 (d, 8.0)	134.4	7.51 (d, 7.6)	134.9	7.45 (d, 7.6)	134.0
7	3.03 (dd, 14.0, 4.8) 2.90 (dd, 14.0, 8.4)	39.7	3.04 (dd, 13.6, 4.0) 2.81 (dd, 13.6, 8.4)	40.3	3.02 (dd, 13.6, 4.8) 2.90 (dd, 13.6, 8.0)	39.9
8	4.29 (dd, 8.4, 4.8)	71.0	4.06 (m)	71.6	4.28 (dd, 8.0, 4.8)	71.1
9	-	173.8	-	175.6	-	173.9
10	-	166.4	-	166.9	-	167.6
9-OMe	3.62 (s)	51.5	-	-	3.61 (s)	51.5
10-OMe	3.84 (s)	52.1	3.84 (s)	52.5	-	-

^a^ measured in DMSO-*d*_6_ at 400 MHz.

**Table 3 molecules-29-00115-t003:** Anti-inflammatory activities of compounds **1**–**6**.

Compound	IC_50_ (µM)	Compound	IC_50_ (µM)
**1**	28.26 ± 0.15	**4**	33.16 ± 0.18
**2**	6.52 ± 0.23	**5**	7.13 ± 0.16
**3**	35.68 ± 0.17	**6**	6.68 ± 0.22
Hydrocortisone ^a^	5.61 ± 0.12		

^a^ Positive control.

## Data Availability

The authors confirm that the data supporting the findings of this study are available within the article or its Appendix A.

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
