# Peer review of "Plumeriapropionics A–E, Carboxyl-Substituted Phenylpropionic Acid Derivatives with Anti-Inflammatory Activity from Plumeria rubra L."

_molecules, 2023, doi:10.3390/molecules29010115_

Round 1

Reviewer 1 Report (Previous Reviewer 1)

Comments and Suggestions for Authors

Major revision:

Line 16: “Compounds 1-6 were evaluated ………” change in “The inhibitory activities of compounds 1-6 against nitric oxide (NO) production induced by lipopolysaccharide (LPS) were evaluated in vitro using mouse macrophage RAW264.7 cells”

Line 19: delete induced

Line 20: change “These results indicate that the discoveries of these carboxyl-substituted phenylpropionic acid derivatives, isolated from the flowers of P. rubra showing significant anti-inflammatory could be of great importance to the research and development of new natural anti-inflammatory agents” with “These results indicate that the discovery of carboxyl-substituted phenylpropionic acid derivatives, from the flowers of P. rubra, which show significant anti-inflammatory properties, could be of great importance to the research and for the development of novel natural anti-inflammatory agents.

Line 28: put commas after Plumeria and family

Line 32: change “have caused” with “have led to”

Linre 36: change “are often as a folk” with “are often used in folk”

Line 45: after “derivatives” insert “named”

Line 47: change “were identified” with “were clarified”

Line 53: change “in the development” with “for the development”

Figure 1. The numbering of compound 1 is wrong. You need to correct it, and the corresponding adjustments will have to be made when you describe the other compounds.

Line 65: delete “which were”

Line 67: after powder put a stitch. The molecular formula…………was deduced by …..

Line 68: change “high-resolution electrospray” with “high-resolution electrospray ionisation mass spectrometry”

Line 82: change “the HMBC correlations from 1-OMe to C-1” with “the HMBC correlations between 10-OMe and C-3”

Line 84: compounds of cerberic acid B delete “of”

Line 84: completely change the sentence “To confirm the above conclusion, the ECD spectra of 1 and 6 were tested, which showed similar features in the 200–400 nm range (Figure 3)” with “To substantiate the aforementioned conclusion, the ECD spectra of 1 and 6 were analyzed, revealing comparable characteristics within the 200–400 nm range (see Figure 3)”.

Line 85: change “tested, which showed similar features in the 200–400 nm range” with “evaluated, revealing similar characteristics within the 200–400 nm range (Figure 3).

Line 88: change “4” with “5”

Line 95: delete at m/z 327.0878 [M-H] – and put in the round brackets; delete minus sign from formula

Line 101: change “that the other parts of the molecule were the same as those of 1” with “that the remaining parts of the molecule were identical to those observed in compound 1”.

Line 111: delete “plumeriapropionic C”

Line 114: change “2” with “3”

Line 126: delete minus sign from formula

Line 128: delete “for 10-OMe in 3”

Line 143: delete minus sign from formula

Line 145: change “3” with “5”

Line 145: change “This was corroborated by comparing the chemical shift of 5 with compound 3 at C-1“ with “This was confirmed by comparing the chemical shift at C-3 of compound 5 with that of compound 3”.

Line 153: delete “to”

Line 163: add carboxylic “group at C-3”

In all molecular formulas you have to remove the minus and the plus signs

I think you could put a final sentence where you say that the stereochemistry at C-9 was defined to be 9-R based on ECD spectra in comparison with the known compound 6. You can move the spectra in supporting information.

Comments on the Quality of English Language

The quality of English needs to be improved

Author Response

Thank you for your letter and comments concerning our manuscript. We have studied comments carefully and have made correction which we hope meet with approval. The main corrections in the paper and the responds to your comments are as following:

  1. Line 16: “Compounds 1-6 were evaluated ………” change in “The inhibitory activities of compounds 1-6 against nitric oxide (NO) production induced by lipopolysaccharide (LPS) were evaluated in vitro using mouse macrophage RAW264.7 cells”

Response: We have corrected them.

  1. Line 19: delete induced

Response: We have corrected it.

  1. Line 20: change “These results indicate that the discoveries of these carboxyl-substituted phenylpropionic acid derivatives, isolated from the flowers of P. rubra showing significant anti-inflammatory could be of great importance to the research and development of new natural anti-inflammatory agents” with “These results indicate that the discovery of carboxyl-substituted phenylpropionic acid derivatives, from the flowers of P. rubra, which show significant anti-inflammatory properties, could be of great importance to the research and for the development of novel natural anti-inflammatory agents.

Response: We have corrected them.

Line 28: put commas after Plumeria and family

Response: We have corrected them.

Line 32: change “have caused” with “have led to”

Response: We have corrected it.

Linre 36: change “are often as a folk” with “are often used in folk”

Response: We have corrected it.

Line 45: after “derivatives” insert “named”

Response: We have corrected it.

Line 47: change “were identified” with “were clarified”

Response: We have corrected it.

Line 53: change “in the development” with “for the development”

Response: We have corrected it.

Figure 1. The numbering of compound 1 is wrong. You need to correct it, and the corresponding adjustments will have to be made when you describe the other compounds.

Response: We have corrected them.

Line 65: delete “which were”

Response: We have corrected it.

Line 67: after powder put a stitch. The molecular formula…………was deduced by …..

Response: We have corrected them.

Line 68: change “high-resolution electrospray” with “high-resolution electrospray ionisation mass spectrometry”

Response: We have corrected it.

Line 82: change “the HMBC correlations from 1-OMe to C-1” with “the HMBC correlations between 10-OMe and C-3”

Response: We have corrected them.

Line 84: compounds of cerberic acid B delete “of”

Response: We have corrected it.

Line 84: completely change the sentence “To confirm the above conclusion, the ECD spectra of 1 and 6 were tested, which showed similar features in the 200–400 nm range (Figure 3)” with “To substantiate the aforementioned conclusion, the ECD spectra of 1 and 6 were analyzed, revealing comparable characteristics within the 200–400 nm range (see Figure 3)”.

Response: We have corrected them.

Line 85: change “tested, which showed similar features in the 200–400 nm range” with “evaluated, revealing similar characteristics within the 200–400 nm range (Figure 3).

Response: We have corrected them.

Line 88: change “4” with “5”

Response: We have corrected it.

Line 95: delete at m/z 327.0878 [M-H] – and put in the round brackets; delete minus sign from formula

Response: We have corrected them.

Line 101: change “that the other parts of the molecule were the same as those of 1” with “that the remaining parts of the molecule were identical to those observed in compound 1”.

Response: We have corrected them.

Line 111: delete “plumeriapropionic C”

Response: We have corrected it.

Line 114: change “2” with “3”

Response: We have corrected it.

Line 126: delete minus sign from formula

Response: We have corrected it.

Line 128: delete “for 10-OMe in 3”

Response: We have corrected it.

Line 143: delete minus sign from formula

Response: We have corrected it.

Line 145: change “3” with “5”

Line 145: change “This was corroborated by comparing the chemical shift of 5 with compound 3 at C-1“ with “This was confirmed by comparing the chemical shift at C-3 of compound 5 with that of compound 3”.

Response: We have corrected them.

Line 153: delete “to”

Response: We have corrected it.

Line 163: add carboxylic “group at C-3”

Response: We have corrected it.

In all molecular formulas you have to remove the minus and the plus signs

Response: We have corrected them.

I think you could put a final sentence where you say that the stereochemistry at C-9 was defined to be 9-R based on ECD spectra in comparison with the known compound 6. You can move the spectra in supporting information.

Response: We have corrected them. Thank you!

Reviewer 2 Report (Previous Reviewer 2)

Comments and Suggestions for Authors

     The revised version of manuscript still requires improvement. There remain too many careless errors. Moreover, this reviewer again recommends that the authors entrust proofreading to a native speaker or an appropriate editing organization. Some issues are listed as follows. The authors should check not only the following points but overall manuscript very carefully.

(1) Line 37: enteritidis –> enteritis??

(2) Lines 46–47: “Structure elucidation” will not be “identified”. Revise this sentence.

(3) Line 68: “HRESIMS” is not the abbreviation of “high-resolution electrospray”. Insert mass spectrometry.

(4) Line 69: 135EDPT –> 135DEPT

(5) Lines 79–80: HMBCs –> HMBC correlations

(6) Line 82: HMBC correlations –> HMBC correlation

(7) Line 83–84: The absolute configuration cannot be “assigned” by comparing the optical rotation of two different molecules. Please revise this sentence.

(8) Line 84: salvianicacid A –> salvianolic acid A??

(9) Line 89: 14 –> 15

(10) Line 108: 12 –> 1 and 2

(11) Line 153–155: In addition to –> In addition, ??

(12) Section 3.1: An equipment for measurement of CD spectra should be described.

(13) Line 192: Frs. 3 –> Fr. 3

(14) Line 196: Fraction Fr. 3-2 –> Fr. 3-2

(15) Lines 203–217: CD spectroscopic data for new compounds 15 should be described.

(16) Lines 230–231: The phrase “further addition to diverse” is not understandable and requires rephrasing.

(17) Figures 3–7 can be transferred to supplementary data.

Comments on the Quality of English Language

There remain too many errors. This reviewer again recommends that the authors entrust proofreading to a native speaker or an appropriate editing organization. 

Author Response

Thank you for your letter and comments concerning our manuscript. We have studied comments carefully and have made correction which we hope meet with approval. The main corrections in the paper and the responds to your comments are as following:

  • Line 37: enteritidis –> enteritis??

Response: We have corrected it.

(2) Lines 46–47: “Structure elucidation” will not be “identified”. Revise this sentence.

Response: We have corrected it.

(3) Line 68: “HRESIMS” is not the abbreviation of “high-resolution electrospray”. Insert mass spectrometry.

Response: We have corrected it.

(4) Line 69: 135EDPT –> 135DEPT

Response: We have corrected it.

(5) Lines 79–80: HMBCs –> HMBC correlations

Response: We have corrected them.

(6) Line 82: HMBC correlations –> HMBC correlation

Response: We have corrected it.

(7) Line 83–84: The absolute configuration cannot be “assigned” by comparing the optical rotation of two different molecules. Please revise this sentence.

Response: We have corrected them.

(8) Line 84: salvianicacid A –> salvianolic acid A??

Response: We have corrected it.

(9) Line 89: 14 –> 15

Response: We have corrected it.

(10) Line 108: 12 –> 1 and 2

Response: We have corrected it.

(11) Line 153–155: In addition to –> In addition, ??

Response: We have corrected them.

(12) Section 3.1: An equipment for measurement of CD spectra should be described.

Response: We have added it.

(13) Line 192: Frs. 3 –> Fr. 3

Response: We have corrected it.

(14) Line 196: Fraction Fr. 3-2 –> Fr. 3-2

Response: We have corrected it.

(15) Lines 203–217: CD spectroscopic data for new compounds 15 should be described.

Response: We have added them.

(16) Lines 230–231: The phrase “further addition to diverse” is not understandable and requires rephrasing.

Response: We have corrected them.

(17) Figures 3–7 can be transferred to supplementary data.

Response: We have corrected them. Thank you!

Round 2

Reviewer 1 Report (Previous Reviewer 1)

Comments and Suggestions for Authors

The compounds must be numbered as reported for cerberic acid, see reference 13.

Comments on the Quality of English Language

English can be improved 

Author Response

Thank you for your letter and comments concerning our manuscript. We have studied comments carefully and have made correction which we hope meet with approval. The main corrections in the paper and the responds to your comments are as following:

  1. The compounds must be numbered as reported for cerberic acid, see reference 13.

Response: We have corrected them. Thank you!

This manuscript is a resubmission of an earlier submission. The following is a list of the peer review reports and author responses from that submission.

Round 1

Reviewer 1 Report

Comments and Suggestions for Authors

This manuscript deals with isolation, structure determination and pharmacological evaluation of five new Carboxyl-substituted Phenylpropionic Acid from Plumeria rubra L. From my point of view this manuscript is not acceptable for Molecules.

The determination of the planar structure of the new compounds is not adequately described. The molecule's numbering does not match the description in the text, leading to considerable confusion. Also, the assignment of the single stereogenic center is not clear. It is stated to be at C9 when the stereogenic center is at C8. 

From my point of view this manuscript is not acceptable for Molecules. The paper lacks in originality and also bears a number of mistakes in spelling and in the use of English language. There are conceptual errors. The determination of the planar structure of compounds 1-5 appeared totally confusing. The NMR data of compounds don’t match those reported in tables 1 and 2. In particular, there are serious concerns about the structural assignment. It is necessary to carefully check all the evidence for structural elucidation and this may necessitate additional complementary analyses of the compounds such as the determination of the configuration at C9.
The sentence (line 83 page 2) “the stereostructure of C-9 was assigned as R-form by comparing the optical rotation of 1 with the known compounds” is completely wrong. C-9 is a quaternary carbon and it is not possible to assign the configuration comparing the optical activity of two different molecules.
Comments on the Quality of English Language

The quality of english language is poor

Reviewer 2 Report

Comments and Suggestions for Authors

     This manuscript describes isolation and structure elucidation of five previously undescribed 3-(3-carboxyphenyl)propionic acid derivatives from the flowers of Plumeria rubra. Inhibitory effects of the isolated compounds on LPS-induced NO production in RAW264.7 cells were also evaluated. The following points require reconsideration.

(1) In determination of the absolute configuration of compounds 15, it is desirable to convert them into compound 6 and then compare the specific rotation of it with that of natural 6. Alternatively, comparison of the experimental and calculated ECD spectra of compounds 15 will provide strong evidence of the absolute configuration.

(2) Position numberings shown in Results section and Tables 1 and 2 do not correspond with those in Figure 1. 

(3) Line 119: NMR data of a methoxy group requires revision.

(4) Lines 200 and 203: The formulae of pseudo-molecular ions for compounds 4 and 5 should be corrected.

Comments on the Quality of English Language

Overall manuscript should be prepared more carefully because there are too many errors in words, spelling, text style, and so on. It is also strongly recommended to entrust proofreading to a native speaker or an appropriate editing organization.